# Application of Box-Behnken Design in the Preparation, Optimization, and In-Vivo Pharmacokinetic Evaluation of Oral Tadalafil-Loaded Niosomal Film

**DOI:** 10.3390/pharmaceutics15010173

**Published:** 2023-01-03

**Authors:** Kawthar K. Abla, Amina T. Mneimneh, Ahmed N. Allam, Mohammed M. Mehanna

**Affiliations:** 1Pharmaceutical Nanotechnology Research Lab, Faculty of Pharmacy, Beirut Arab University, Beirut, Lebanon; 2Department of Pharmaceutics, Faculty of Pharmacy, Alexandria University, Alexandria 21521, Egypt; 3Department of Industrial Pharmacy, Faculty of Pharmacy, Alexandria University, Alexandria 21521, Egypt

**Keywords:** Box-Behnken, methylcellulose, niosomes, oral film, tadalafil

## Abstract

Benign prostatic hyperplasia (BPH) affects about 90% of men whose ages are over 65. Tadalafil, a selective PDE-5 inhibitor, was approved by FDA for BPH, however, its poor aqueous solubility and bioavailability are considered major drawbacks. This work intended to develop and evaluate oral fast dissolving film containing tadalafil-loaded niosomes for those who cannot receive the oral dosage form. Niosomes were statistically optimized by Box-Behnken experimental design and loaded into a polymeric oral film. Niosomes were assessed for their vesicular size, uniformity, and zeta potential. The thickness, content uniformity, folding endurance, tensile strength, disintegration time, and surface morphology were evaluated for the prepared polymeric film. The optimized niosomes revealed high entrapment efficiency (99.78 ± 2.132%) and the film was smooth with good flexibility and convenient thickness (110 ± 10 µm). A fast release of tadalafil was achieved within 5 min significantly faster than the niosomes-free drug film. The in-vivo bioavailability in rats established that the optimized niosomal film enhanced tadalafil systemic absorption, with higher peak concentration (C_max_ = 0.63 ± 0.03 µg/mL), shorter T_max_ value (0.66-fold), and relative bioavailability of 118.4% compared to the marketed tablet. These results propose that the oral film of tadalafil-loaded niosomes is a suitable therapeutic application that can be passed with ease to geriatric patients who suffer from BPH.

## 1. Introduction

Benign prostate hyperplasia (BPH) is one of the most common diseases in aging men and the most common cause of lower urinary tract symptoms (LUTS). Clinically, BPH arises from the tension exerted by the prostate smooth muscle on the prostatic urethra and nearly every man age 65 or older will be affected by it at some point in his life [1]. In addition, many patients do not stand the conventional medications; alpha-blockers and 5-alpha reductase inhibitors, for their adverse effects such as dizziness, postural hypotension, reduced ejaculation, loss of libido, and male infertility [2,3] which creates the need for another molecule. Clinical studies have granted evidence that PDE-5 inhibitors enhance BPH and LUTS symptoms due to the relaxing action of nitric oxide (NO) and the inhibition of prostatic stromal cells proliferation [4,5].

Tadalafil was approved in 2003 by US Food and Drug Administration (FDA) as the first and only PDE5 inhibitor clinically proven to provide sustained efficacy and in 2011, a single 5 mg daily dose was licensed to treat LUTS secondary to BPH with or without erectile dysfunction. Pharmacokinetically, it is known for its long half-life and duration of action (17.5 and 36 h, respectively) in comparison to sildenafil (3.8 and 8 h) and vardenafil (3.9 and 12 h) [6]. Despite the previously mentioned ascendancy, one of the demerits that hinders tadalafil action is its poor bioavailability which was found to be around 35% of the administered dose [7,8]. This is attributed to its low aqueous solubility since it belongs to class II drugs according to the FDA biopharmaceutical classification system [9].

Nanotechnology has emerged as a successful approach to the creation of nanocarriers for distinct diseases [10]. Conventional therapeutics are somehow destined to failure due to problems related to solubility, bioavailability, distribution, or transport through biological membranes. On the other hand, nanocarriers can be designed in a way that leads to manipulative targeting, optimal delivery, and the release of drugs to achieve the desired outcomes [11,12,13]. Several strategies in the nanotechnology were adopted to enhance tadalafil solubility and dissolution rate [14,15,16,17,18]. Nevertheless, the niosomal approach to enhance tadalafil solubility has not been studied before despite all the merits that characterized this nanoparticulate system.

Niosomes are bi-layered nanocarriers that can be prepared by the addition of non-ionic surfactant to cholesterol in a proper proportion followed by its hydration in aqueous media [19]. Niosomes possess many advantages over other nanocarriers due to their high biocompatibility, biodegradability, high stability, and ease of surface modification. In the bargain is their ability to encapsulate both hydrophilic and hydrophobic drugs in a special geometrical structure thus enhancing the bioavailability of certain agents [20]. It was reported by Aboubaker et al. that glutathione niosomes remarkably improved drug oral bioavailability and hepatic tissue uptake [21].

In any pharmaceutical dosage form, the ultimate goals are to improve patient compliance along with the rapid onset of action, convenient handling, and ease of administration. Oral films have attracted expanding interest in pharmaceutical technology in the last few years due to their supremacies. The use of an orally dissolving, permeable film delivery system can serve as a more efficient route to deliver tadalafil. Oral films are flexible and easy to use, making them a more satisfactory and convenient dosage form when compared to conventional oral dosage forms which advocate patient compliance, specifically for children and elderly patients, or patients with swallowing dysfunction or dysphagia where complete and accurate dosing can be difficult to attain [22,23]. Visser et al. showed that orodispersible films were the most favorite candidates for patients who were nursing home residents [24]. Besides, films can be utilized to overcome bioavailability problems of drugs that are vulnerable to poor aqueous solubility by promoting oromucosal absorption, directly entering the systemic circulation [25].

Moreover, the consolidation of both niosomes and film in one dosage form can provide a lot of merits in enhancing therapeutic outcomes. Allam and Fetih, for example, stated that loading metoprolol tartrate in a niosomal oral film enhanced significantly its bioavailability compared to the marketed tablet dosage form and produced a prolonged effect of the drug with no detected side effects [26]. Herein, combining the features of niosomes with the oral film increases the drug solubility of poorly soluble drugs and creates a formulation with reduced local irritation, uniform dispersion in the targeting site, more reproducible drug absorption, and enhanced bioavailability [27].

In the current piece of work, an attempt has been made to formulate and assess oral film containing tadalafil-loaded niosomes to provide a fast systemic effect by improving tadalafil solubility through a patient-geriatric friendly formulation for those suffering from benign prostatic hyperplasia. Box-Behnken design was purposed to optimize the niosomal formulation. The optimized film containing niosomes was evaluated for its physical and mechanical properties, including surface morphology. Besides, film pharmacokinetics characteristics were appraised in rats against the marketed product.

## 2. Materials and Methods

Tadalafil (≥98%), polyethylene glycol 400 (PEG 400) (~98%), microcrystalline cellulose EP (Avicel^®^ PH-101), methylcellulose (MC) (~98%), polyethylene-polypropylene glycol (Poloxamer^®^407), polysorbate 80 (Tween^®^80), sorbitan monostearate (Span^®^60), saccharine sodium, and menthol were purchased from Sigma-Aldrich, Steinheim, Switzerland. Fat-free soybean phospholipids with 70% phosphatidylcholine (Lipoid S^®^75) were kindly gifted from Lipoid AG, Sennweidstrasse, Switzerland. All other reagents and solvents used were of analytical grade.

### 2.1. Preparation and Statistical Optimization of Tadalafil-Loaded Nanovesicles

#### 2.1.1. Preparation of Tadalafil-Loaded Niosomes

The conventional ether-injection method was applied for the screening of non-ionic surfactants and the preparation of the drug-loaded niosomes as Bendas et al. with modifications [28]. Accurately weighed quantities of three nonionic surfactants (span^®^60, tween^®^80, and poloxamer^®^407) were mixed with lipoid in a 2:1 molar ratio and dissolved in diethyl ether. The resulting solution was injected using a microsyringe at a rate of 1 mL/min into a preheated distilled water which was maintained at 65 °C temperature under stirring until complete solvent removal. The surfactant that produced niosomes with the smallest size, minimal polydispersity index, and optimal zeta potential was selected to be used for niosomes formulation.

The drug-loaded niosomes were prepared by the same procedure where the etheric solution of nonionic surfactant and Lipoid S^®^75 was mixed with methanol formerly containing the required amount of tadalafil.

#### 2.1.2. Box–Behnken Design (BBD) Experiment

The tadalafil-loaded niosomes were optimized using a Box-Behnken response surface methodology experimental design (Design-Expert^®^ Software Version 11) (3 factors, 3 levels). The independent variables selected were mixing time (X1), non-ionic surfactant to lipid ratio (X2), and the total weight of the preparation components (X3) with their low, medium, and high levels for preparing 15 formulations as given in Table 1. The studied responses were particle size (Y1), polydispersity index (Y2), and zeta potential (Y3). Moreover, 3D response surface graphs were plotted to depict the effects of the predetermined factors on the responses measured.

#### 2.1.3. Particle Size, Polydispersity Index, and Zeta Potential

The surface potential, vesicle size, and polydispersity index (PDI) of tadalafil-loaded niosomes were measured at 25 ± 2 °C and 90° scattering angle by Zeta sizer 2000 (Malvern Instruments, England, UK) [15]. The above-mentioned characteristics were measured to aid further in the optimization of the preparation. All measurements were carried out in triplicate, the mean and the standard deviation were computed.

#### 2.1.4. Entrapment Efficiency

Niosomes containing tadalafil were separated from the unloaded free drug by applying Mehanna et al. separation technique [29]. In this indirect method of separation, cooling centrifugation at 15,000 rpm for 60 min at 4 °C was executed (Sigma 3–30KS centrifugation, Osterode, Germany). The supernatant was separated each time, then combined and assayed spectrophotometrically at 291 nm. The amount of entrapped drugs was attained by deducting the amount of free drugs from the total added drug. The percent of entrapment efficiency (EE %) was then calculated as Equation (1). The results are expressed as the mean of three separate experiments:(1)$$\mathrm{EE}\%=\frac{\;\mathrm{Total}\;\mathrm{amount}\;\mathrm{of}\;\mathrm{drug}-\mathrm{amount}\;\mathrm{of}\;\mathrm{free}\;\mathrm{drug}}{\mathrm{Total}\;\mathrm{amount}\;\mathrm{of}\;\mathrm{drug}\;}\times 100$$

### 2.2. Preparation and Characterization of Tadalafil-Loaded Noisomal Fast Dissolving Films

#### 2.2.1. Preparation of Tadalafil-Loaded Niosomal Oral Film

Based on literature, methylcellulose (MC) was tested as film-forming material for the preparation of fast dissolving film [26]. Polyethylene glycol 400 was used as a plasticizer, saccharine as a sweetener, and menthol as a flavoring agent. Microcrystalline cellulose MCC (Avicel^®^) was used as a superdisintegrant owing to its short disintegration time [26]. The solvent casting method was applied for the preparation of the film. Initially, the polymer was dispersed in the casting vehicle (distilled water), followed by the addition of saccharine and menthol. Furthermore, MCC (Avicel^®^) was levigated with PEG 400 before amalgamation into the polymeric solution. After a volume of tadalafil-loaded niosomes (corresponding to the required dose) was incorporated and mildly mixed with the polymeric solution and the final volume was adjusted to 25 mL with distilled water. Later, the solution was poured on the petri dish (surface area, 19.63 cm^2^) and allowed to dry until a constant weight. The patches were cut into 4 cm^2^ pieces and stored in a dry place at room temperature to preserve their integrity and elasticity.

#### 2.2.2. Vesicle Size Analysis of the Niosomal Film

Vesicle size of the niosomal film was analyzed after reconstitution. The film was dispersed in deionized water followed by sonication for 10 min. The obtained dispersion was analyzed at 25 ± 2 °C and 90° scattering angle for the determination of average particle size, polydispersity index, and zeta potentials for the reconstituted vesicles using Zeta sizer 2000 (Malvern Instruments, England, UK) [30].

#### 2.2.3. In-Vitro Assessment

The niosomal oral film was characterized for its physical appearance, thickness, weight, and content uniformity. Moreover, surface pH, moisture content, folding endurance, tensile strength, disintegration time, and surface morphology of the prepared polymeric film were evaluated.

The physical appearance was checked via visual inspection of the films. The thickness of the films was measured by a micrometer screw gauge at five different points, followed by calculating their mean value [31].

Weight variation test was performed by determining individual weights of 4 cm^2^ pieces from ten randomly selected films using an analytical balance (MC-1 AC210S, Sartorius, Goettingen, Germany) to calculate the average weight and the standard deviation [32].

Content uniformity in the formulated films was computed by dissolving 10 randomly selected in 100 mL phosphate buffer (pH 6.8). The resulting solutions were then filtered, and further dilution was carried out with phosphate buffer to measure their absorbance spectrophotometrically at 291 nm to determine tadalafil content [33].

For determining the surface pH, the niosomal films were allowed to be in contact with distilled water and then measured by a pH digital meter (SED 12,500 V Martini Instruments Co., Ltd., Beijing, China) at room temperature (25 ± 0.2 °C) by bringing the electrode in contact with the film surface and allowing them to equilibrate for about one minute [34].

Percentage moisture content was measured by weighing the films before and after drying them in a hot air oven at 105 ± 5 °C for 2 h to indicate the difference in weight according to the following equation [35]:(2)$$\%\;\mathrm{Moisture}\;\mathrm{content}=\frac{\mathrm{Initial}\;\mathrm{weigh}-\mathrm{final}\;\mathrm{weight}}{\mathrm{initial}\;\mathrm{weight}\;}\times 100$$

The mechanical strength was determined through the folding endurance of the prepared films which was computed based on the number of times the film can fold at the same place without breaking [36].

The tensile strength of the prepared oral film was measured using a digital tensile testing machine (Lloyd Instruments Ltd., LR 10K, Bognor Regis, UK). The film was cut into a rectangular shape and the tensile tester was set with a crosshead speed of 35 mm/minute and an initial grip separation of 25 mm. The film was placed vertically between the tensile tester’s two clamps which they were pulled apart until the film broke [27]. Tensile strength was calculated as follows [37]:(3)$$\mathrm{Tensile}\;\mathrm{strength}=\frac{\mathrm{Force}\;\mathrm{at}\;\mathrm{breakge}\;\left(\mathrm{kg}\right)}{\mathrm{Area}\;\left({\mathrm{mm}}^{2}\right)}\;$$

Moreover, the strain (% elongation) was evaluated to detect the stretching and toughness of the prepared polymeric film. The film elongation before breakage is referred to as the strain or percentage elongation which was determined as follows [35]:(4)$$\%\;\mathrm{Elongation}=\frac{\mathrm{Increase}\;\mathrm{in}\;\mathrm{the}\;\mathrm{film}\;\mathrm{length}}{\mathrm{Initial}\;\mathrm{length}\;\mathrm{of}\;\mathrm{film}\;}\times 100\;$$

In-vitro disintegration time was calculated by placing the film into a petri dish containing phosphate buffer (pH 6.8) and swirling it every 10 s. The disintegration time was computed when the film started to disintegrate or break down [26].

Scanning electron microscopy (SEM-JEOL, JSM-840A, Takamatsu, Japan) examined the film surface topography and morphology. The films were mounted on the sample stab using a double-sided adhesive tape and were coated with gold (200A) under reduced pressure for 5 min to enhance the conductivity using an ion sputtering device (JEOL, JFC-1100E, Takamatsu, Japan) [38].

#### 2.2.4. In-Vitro Release Study

The release pattern of tadalafil from the niosomal film was conducted using the beaker method with slight modifications to compare free-drug-containing film [39,40]. A solution of 0.1 N HCl and 2% *w*/*v* tween^®^80 (pH 1.2) was used as a dissolution medium as it was previously proved to be a compatible medium for the release of tadalafil [15]. The dissolution medium was placed in a shaking water bath (FALC, WB-MF24, Treviglio, BG, Italy) set at 37 ± 0.5 °C and 100 rpm. Samples were periodically withdrawn at different time intervals and assayed using a UV-visible spectrophotometer (Jasco V-730 spectrophotometer, Heckmondwike, UK) at 291 nm. Samples were replaced with the same volume of fresh dissolution medium to maintain sink condition. The experiment was conducted in triplicates. The amount of drug released at each time interval was calculated, and the cumulative amount of drug released was determined as a function of time to construct the drug release profile graph. The release kinetics of tadalafil-loaded niosomal film was investigated by the curve fitting method to different mathematical models [41].

### 2.3. In-Vivo Assessment

#### 2.3.1. Experimental Animals

Male albino Wistar rats weighed between 130–150 g were used for the in-vivo pharmacokinetic study. Rats were obtained from the animal house of the Faculty of Pharmacy, Beirut Arab University (BAU), Lebanon. Rats were housed in polyacrylic cages under standard animal housing conditions before and during the experiment. The animals had access to water and standard laboratory chow.

Animal handling during the work was carried out following the regulations and guidelines stipulated by the Institutional Animal Care and Use Guidelines (IACUG) at BAU, Lebanon and authenticated by the Ministry of Public Health. All experiments were performed at Beirut Arab University laboratories after obtaining approval from the Investigation Review Board (IRB), number: 2022-0045-P-M-93.

#### 2.3.2. In-Vivo Pharmacokinetic Study

Tadalafil-loaded niosomal oral film and the marketed tablet were evaluated for their pharmacokinetic parameters in male albino Wistar rats. Randomly, the rats were divided into two groups, each of five rats. The first group received the oral film and the second received the marketed tablet. The animals of the first group were pre-anesthetized with ether where a film containing 5 mg of tadalafil was placed in the rat buccal cavity with the help of forceps and a Teflon spatula. The second group received an aqueous oral suspension of the marketed tablet using the gavage technique. Blood samples were withdrawn from the tail vein at predetermined time intervals post-dose. The blood samples were collected in heparinized tubes and subjected to centrifugation (Centurion Scientific, Chichester, UK) at 4500 rpm for 15 min at 4 ± 0.2 °C. The plasma was carefully separated using a micropipette and stored in Eppendorf tubes at −80 ± 5 °C (So-Low, Ultra-Low Freezer, Environmental Equipment, Cincinnati, OH, USA) for further analysis [37].

#### 2.3.3. Extraction of Tadalafil from Plasma

The frozen plasma sample was thawed at room temperature (25 ± 2 °C) for the tadalafil quantification assay. Acetonitrile was added to 100 µL of the thawed plasma sample to precipitate the proteins followed by vortexing the mixture for 5 min. The sample was then centrifuged at 8500 rpm at 4 ± 0.2 °C for 20 min from which 100 µL of the clear supernatant was transferred into a clean vial, filtrated, degassed, and analyzed using reverse-phase high-performance liquid chromatography [42].

#### 2.3.4. HPLC Analysis

Quantitative analysis was carried out using an Agilent technology (Waldbronn, Germany) equipped with C18 column (250 × 4.6 mm LD), autosampler, pump, and photodiode array detector. An isocratic mobile phase consisting of buffer (potassium dihydrogen orthophosphate) and acetonitrile (1:1 *v/v*) mixture was eluted at a flow rate of 1.2 mL/min at ambient temperature (25 ± 0.5 °C) and the column effluent was detected at 285 nm wavelength with a total run time of 18 min. Tadalafil retention time was found to be 3.18 min. Tadalafil concentration was determined through a calibration curve of plotted peak area versus concentration [43].

#### 2.3.5. Pharmacokinetic Parameters

Tadalafil pharmacokinetics calculations were performed using a non-compartmental approach, applying Kinetica^®^ 4.4.1 SPSS 14 software^®^. Tadalafil concentration-time curve was used to obtain the tadalafil maximum concentration (C_max_, µg/mL) and the time of tadalafil maximum concentration (T_max_, hours). The area under the concentration-time curve from zero to the last analyzed point (AUC_0–24_, µg.h/mL) was computed using the linear trapezoidal rule and cp/k was added to obtain AUC_0–∞,_ where cp is the last measured concentration and k is the elimination rate constant. T_1/2_ (hour) was determined based on the first order. Fraction (F) of drug absorbed and relative bioavailability were calculated according to the following equations:(5)$$F=\frac{\mathrm{AUC}\;*\;\mathrm{clearance}\;\left(\mathrm{CL}\right)}{\mathrm{Amount}\;\mathrm{of}\;\mathrm{drug}\;\mathrm{dose}\;\left(x\right)}$$
(6)$$\mathrm{RB}=\frac{\mathrm{AUC}\;0-t\;\left(\mathrm{film}\right)}{\mathrm{AUC}\;0-t\;\left(\mathrm{reference}\right)}\times 100$$

### 2.4. Stability Study

The stability evaluation of the optimized tadalafil-loaded niosomal film was assessed at two distinct storage conditions. Patches of the film were stored in aluminum packages at 25 °C with 50–60% humidity (ordinary conditions) and 40 °C with 75% humidity (accelerated conditions) for twelve weeks, then the content of tadalafil was determined, in addition to other parameters, as weight and thickness [44].

### 2.5. Statistical Analysis

The obtained results of the two treated groups were expressed as mean ± standard deviation (SD) and compared using a one-way analysis of variance or a two-sided Student’s *t*-test for pairwise comparison where results were considered statistically significant if the *p*-value was ≤0.05.

## 3. Results and Discussion

### 3.1. Box-Behnken Statistical Optimization of Tadalafil-Loaded Niosomal Formulation

Upon applying the ether injection method in the preparation of niosmes, the alteration in temperature between phases governed by the slow injection of the lipid component into the aqueous phase encouraged the swift vaporization of solvent, ensuring spontaneous vesiculation and formation of niosomes [45]. The pre-formulation studies revealed that tween^®^80 was a suitable non-ionic surfactant as it aided in the production of niosomes with small particle size (147 ± 2.63 nm), PDI (0.331), and ideal zeta potential (−31.3 ± 0.87 mV) compared to poloxamer^®^407 and span^®^60 (Table 2). The incorporation of tween^®^ in the lipid bilayers of the vesicles provided control over the shape, size, phase transition temperature, and fluidity of the niosomal vesicle [46]. This result is analogous to Alyami et al. where the particle size of niosomes containing tweens^®^ was smaller than those containing Spans^®^. The hydrophilic head group of tweens^®^ led to the formation of a thin bilayer and thus a smaller particle size [47].

The present study optimized the preparation of niosomes via ether injection method by applying the Box-Behnken design. Each independent factor was examined at three levels, in addition to their binary interactions and their polynomial effects. The inspected features of the prepared systems were the vesicular size, polydispersity index, and surface charge which are depicted in Table 3. Cubic mathematical models were applied to analyze the relationship between the independent factors and the studied responses (Equations (7)–(9)):Y_1_ = +156.55 − 10.935 A + 29.359 B + 9.232 C + 7.390 AB + 9.925 AC + 8.463 BC − 13.476 A^2^ + 55.086 B^2^ + 5.701 C^2^
(7)

Y_2_ = +0.302 − 0.017 A + 0.203 B + 0.016 C + 0.017 AB + 0.016 AC − 0.044 BC − 0.030 A^2^ + 0.178 B^2^ + 0.027 C^2^
(8)

Y_3_ = −29.37 + 0.793 A − 3.99 B + 0.343 C − 0.205 AB + 0.800 AC − 0.320 BC + 1.036 A^2^ − 6.689 B^2^ + 0.741 C^2^
(9)


As presented in Table 3, the obtained data proposed a quadratic model for the analysis of particle size, PDI, and zeta potential, and the difference between adjusted and predicted R^2^ values for the investigated responses was less than 0.2 which indicates a reasonable agreement in the study design. The particle size ranged from 126.80 nm (F14) and 267.70 nm (F6), polydispersity index from 0.224 (F14) and 0.760 (F3) and the zeta potential from −27.80 mV (F14) and −40.00 mV (F3) (Table 4).

From the studied independent variables, the total weight of the formulation (X_3_) showed minimal correlation and didn’t affect the investigated parameters, viz. particle size, PDI, and zeta potential, whether it was at its minimum or maximum the value of the formulation total weight did not possess any effect.

The *p*-values computed in this study were less than 0.05, thus revealing the significant influence of the formulation variables, specifically the time of mixing and surfactant to lipid ratio on the studied responses. The contour plots shown in Figure 1A,B illustrates that upon prolongation of mixing time from 15 to 30 min, both the niosomes particle size and polydispersity index decreased resulting in a more uniform vesicles size distribution. This was parallel to what was reported by Shah et al. where increasing mixing times from 30 to 60 min ensured lower particle sizes and PDI. This comportment may be elucidated by the fact that the short time for mixing is not satisfactory to form complete uniform niosomes [48]. Concerning tween^®^80 to lipoid^®^ ratio, it was observed that as the non-ionic surfactant increased in the formula composition, the formulation had a larger particle size as in F6, F10, and F11 (Table 4, Figure 1A). The presence of a surfactant with high HLB induces niosomal vesicle size enlargement as the surface free energy reduces upon increasing surfactants hydrophobicity [49]. Howbeit, particle size approached adequate values when an appropriate ratio was optimized (1:0.8) as shown in formulations F5, F14, and F15 with a particle size of 150.90, 126.80, and 155.90 nm, respectively (Table 4).

The zeta potential of the runs ranged between −27.80 mV (F14) and −40.00 mV (F3). These high negatively charged surface particles reflected positively on the stability of the niosomes where the vesicles will tend to repel rather than aggregate. Okore et al. proposed a characteristic line to separate between stable and unstable vesicles, which was roughly taken at either +30 or −30 mV [50]. Sailaja et al. attained kindred outcomes with the preparation of naproxen-loaded niosomes by the ether injection method, where tween^®^80 was considered the suitable surfactant for such formulation with a zeta potential of −31.9 mV, suggesting its stability [51].

After experimentally executing the different 15 runs, formula (F14) with the smallest particle size (126.80 nm), lowest PDI (0.224), and optimal zeta potential (−27.80 mV) was selected. This formula was obtained with tween^®^80 to lipoid^®^ molar ratio of 1:0.8, a total weight of 200 mg, and mixed for 30 min. According to the Box Behnken design, the optimal desirable solution for each of the variables and independent factors is illustrated in Figure 2. The data in Table 5 shows a small residual value between the expected and the observed ones for particle size, polydispersity index, and zeta potential. The measured values mirror close concession between the predicted values and the minimal standardized residuals to imply the validity of the developed mathematical model within this design space to interpret the effect of surfactant to lipid ratio, the weight of the formulation, and mixing time on the formulated tadalafil-loaded niosomes.

### 3.2. Entrapment Efficiency

The entrapment efficiency for the optimized prepared niosomal formulation employing the ether injection method was 99.78 ± 2.13%. The high EE of tadalafil in the niosomal preparation indicates the effectiveness of niosomes in solving the poor aqueous solubility of this drug. Similar outcomes were conquered by Sezgin-Bayindir et al. where the EE of candesartan cilexetil-loaded niosomes was high (99.06 ± 1.74%) proving that niosomes can be applied to enhance the aqueous solubility of these drug candidates [52]. The high EE is pretentious by the chain length and size of the hydrophilic head group of the non-ionic surfactant employed. Nonionic surfactants with stearyl (C18) chains display higher entrapment efficiency than those with lauryl (C12) chains. Here the utilization of tween^®^ bearing a long alkyl chain and a large hydrophilic moiety with lipoid^®^ S75 provided this high entrapment efficiency. The relationship between the chain and the entrapment efficiency is controversial, Ruckmani and Sankar showed that tween^®^80 with a longer saturated alkyl chain than tween^®^60 and tween^®^20 exhibited lower entrapment efficiency, which increased from 79.5 ± 0.8% to 82.4 ± 1.4%, and 83.8 ± 1.2% for the latter, respectively, concluding that the lower the HLB value of the surfactant, the lower the entrapment efficiency [46]. However, these results were divergent from those declared by Ahmed et al. which suggested that the lower the HLB of the surfactant, the higher will be the entrapment efficiency [53].

### 3.3. Vesicle Size Analysis of the Niosomal Film

The average particle size of the reconstituted niosomes was 151 ± 3.09 nm with polydispersity index of 0.341± 0.05, and zeta potential of value 29.9 ± 0.9 mV. The result revealed that there was a non-significant difference in particle size of reconstituted film compared with tadalafil niosomal dispersion (*p* > 0.05). The polydispersity index was narrow indicating a narrow particle size distribution and a good resdispersibility of the film containing tadalafil-loaded niosomes of drug within the nanocarrier in the film [54].

### 3.4. In-Vitro Assessment of Tadalafil-Loaded Niosomal Oral Film

The formulation should be able to form a film with sufficient elasticity, softness, flexibility, and good physicochemical stability. As a result, these parameters should be evaluated carefully during the development of oral films to assure their effective performance. Studying the quality attributes of a film is a prerequisite that includes assessing properties such as surface morphology, weight, thickness and content variation, surface pH, mechanical strength, moisture content, disintegration time, and in-vitro release [55]. Results of the physical and mechanical properties of tadalafil oral film are presented in Table 6.

According to the visual inspection and morphological features, the prepared tadalafil-loaded niosomal film was elegant, transparent, flexible, homogenous, and smooth which indicates the good dispersion of the niosomes within the film and the viability of the preparation method [56].

The film thickness should be evaluated since it is related directly to the quantity of the drug within the film. A suitable thickness is also crucial for the comfortable application of the oral film. The optimal thickness should be between 50 and 1000 µm to be suitable for oral administration [57]. As tadalafil film thickness was around 110 ± 10 µm, it would be convenient for oral use. These results are in agreement with previously reported data on film thickness, which was around 160 µm [58].

Similarly, determining the weight variation of the film is necessary to ensure the consistency of the film preparation, repeatability of the technique, as well as drug uniformity [59]. The average weight of the films was around 5.74 mg with a small standard deviation value (±0.29 mg) signifying the chance for non-uniformity in tadalafil content which was also confirmed by the content uniformity testing.

Content uniformity is performed to determine drug content in the individual oral patch and to check the reproducibility of the technique. Tadalafil content was almost the same among the prepared films ~97.82%. This result was coherent with another study in which the film contained metoclopramide and methylcellulose as a polymer and displayed content uniformity of around 95% [60]. These values are accepted according to USP27 which specifies that the content should be between 85% and 115% with less than 6% standard deviation [61].

A film that has inappropriate pH either toward basic or acidic medium may induce damage to the mucosal layer lining the oral cavity leading to patient discomfort. The mean value of surface pH of tadalafil film was 6.67 ± 0.49, with a non-significant difference (*p* > 0.005), which is within the range of the oral cavity pH (6.4–6.8), thus the film is less likely to be irritant to the oral mucosal membrane [62].

The moisture content affects the friability, brittleness, and stability of oral films. Many factors are responsible for increasing the film moisture level as the solvent system, the drug hygroscopicity, the excipients in the formula, and the manufacturing techniques. The moisture content of the prepared tadalafil-loaded niosmal film was 3.02 ± 0.60% which is considered within the acceptable range (<5%) [63]. This result was analogous to the Linku and Sijimol investigation where the moisture content of the prepared polymeric film varied between 1.1% and 3.84% [64].

Folding endurance is carried out to estimate the mechanical properties of the film. In another word, it is performed to detect the flexibility of the film to ensure it can be administrated without breakage. The film has a great mechanical strength when it requires more than 300 time folding to break and develop visible cracks [65]. The folding endurance of tadalafil film was about 320, indicating good flexibility which is indirectly related to the appropriate concentration and dispersion of the methylcellulose within the fabricated formula.

An ideal oral film should display an adequately high tensile strength to be able to withstand normal handling. Despite this, a very high rigid film is not desired, because it could retard the drug release from the polymer matrix. The prepared niosomal oral film had an average tensile strength of 0.79 ± 0.03 MPa which is in line with the previously prepared polymeric film which has a tensile strength value of 0.78 ± 0.05 MPa [66]. The percentage elongation of the prepared film was 6.66 ± 0.12% which is considered ideal for the polymeric film. These data suggested that polyethylene glycol 400, which was used as a plasticizer, was able to reduce the glass transition temperature of methylcellulose and promoted its plasticity and flexibility. This temperature is one of the most vital properties that determine chain mobility of polymer in which the polymer transforms from hard, glassy material to soft, rubbery material with accepted tensile strength and percentage elongation [67].

The disintegration time is the time needed by the film to disperse or disintegrate when it comes in contact with the saliva. The film thickness and weight affect greatly the physical properties of the film. In general, the disintegration time ranges from 5 to 30 s to allow faster drug release and fast oromucosal absorption [61]. The disintegration time of the prepared film was 30.31 ± 3.64 s, which is considered rapid and acceptable compared with niosomal films prepared by Allam and Fetih, where they showed disintegration times ranging from 38 to 180 s [26].

The surface morphology of the formulated film was examined using SEM to clarify its feature. As illustrated in Figure 3, the film showed a continuous, smooth, and homogenous surface. Besides, niosomes vesicles can be visualized in the niosomal film with spherical and smooth surface without any aggregation and the nanometric size range signifying successful incorporation of the selected niosomes within the optimized film. A similar result was observed in Arafa et al. study where SEM images of the prepared propolis-based oral film showed spherical niosomes and smooth features of the film [68].

### 3.5. In-Vitro Release of Tadalafil from the Optimized Niosomal Film

The percentage of tadalafil released from the formulated niosomal film in comparison with its release from the film without niosomes is illustrated in Figure 4. The drug in the niosomal film demonstrated a fast release within 5 min (22.65 ± 1.432%), significantly greater than that from the film alone (7.72 ± 6.782%, *p* < 0.05). Comparable verdicts were reported by Khan et al. where a burst release of ceftriaxone and rifampicin from all the niosomal formulations was detected at the beginning of the dissolution testing compared with drugs alone [69].

In the present study, the release of tadalafil in both formulations increased with the progress of the study. After 30 min the niosomal film provided 78.95 ± 1.117% release of tadalafil, whereas it reached only 10.21 ± 6.012% from the film (*p* < 0.05). The water solubility of MC utilized created porosity in the film, allowing the surrounding solvent to penetrate the film, thus accelerating its dissolving, similar results to what was attained by Auda et al. [51]. Additionally, employing microcrystalline cellulose (Avicel^®^) added faster disintegration of the formulation and hence rapid initial release. These data were in agreement with those of Serrano et al. as the disintegration time was enhanced when higher amounts of Avicel^®^ were employed [70].

The pattern of release continued in a controlled manner; the tadalafil percentage release was 81.94 ± 0.234% at 45 min and reached 85.76 ± 3.123% at the end of the one-hour study. As for the release from the film without niosomes, it was significantly less with 11.30 ± 2.765 and 13.04 ± 3.665% at 45 and 60 min, respectively (*p* < 0.05). The controlled release template was achieved by the incorporation of the drug into the niosomal structures. Niosomes delay the release of the encapsulated drugs due to the presence of lipids in their composition. The existence of lipids decreases the niosomes membrane fluidity by lessening the leakage and permeability of the drugs [71]. Similar data was endorsed by Shailaja and Shreya, as a formulation containing tween 80 showed an optimal release profile of naproxen from the niosomes with 88.9 ± 0.71% after 12 h [51].

In this study, the oral niosomal film provided a dual release pattern characterized by a fast-dissolving profile at the beginning of the experiment yielded by the film components and a controlled one for the remaining time assured by niosomes.

To investigate the release kinetics of tadalafil from the niosomal film, different mathematical release models were adapted, namely, Higuchi, Hixson-Crowell, Korsmeyer-Peppas, first-order, and zero-order models. From the outcomes, it was noticed that tadalafil release from the niosomal film fitted into the Korsmeyer-Peppas model displaying a linear relationship (R^2^ = 0.9986). The release exponent ‘n’ corresponding to the mechanism of drug release was 0.472, indicating that it falls within the range of Fickian diffusion [72]. These inputs were coherent with those of Sadeghi et al. where the release of lysozymes from the niosomes followed the Korsmeyer-Peppas model and *n* was 0.33, thus quasi-Fickian diffusion determined the drug release mechanism [73]. In this study niosomes as a nanocarrier are considered as s drug reservoir since it was able to control the release of tadalafil governed by the small size of the formulation and lipophilicity of lecithin employed that retarded the release.

### 3.6. In-Vivo Pharmacokinetic Assessment

The plasma concentration versus time curves and pharmacokinetic parameters of tadalafil after a single dose administration (oral film and the marketed tablet) are illustrated in Figure 5 and Table 7. Animals who received tadalafil film orally displayed a plasma level-time profile characterized by significantly higher peak concentration (C_max_) with a larger area under the curves (AUC_0–24_ and AUC_0–∞_) compared with those that received the marketed tablets (*p* ≤ 0.05). Additionally, at studied time points, the mean tadalafil plasma concentrations were higher in rats treated with the investigated formula than in those treated with the marketed tablet. Moreover, tadalafil-loaded niosomal film exhibited a significantly shorter T_max_ value (0.66 fold) compared to the marketed tablet (*p* < 0.05). The relative bioavailability of the formulation upon comparison with the marketed tablet was 118.4%, indicating that the oral film improved tadalafil bioavailability.

These findings can be credited to many reasons, first, the presence of tadalafil within the niosomes as a nanocarrier enhanced its solubility as well as its dissolution rate compared to the marketed product [35]. Furtherly, the rapid disintegration of the niosomal oral film in the saliva caused faster tadalafil absorption via the buccal mucosa, therefore reaching higher plasma concentrations more rapidly than that in the control group through pre-gastric absorption. Furtherly, it is worth mentioning that oral mucosa is highly vascularized, which helps in rapidly achieving tadalafil therapeutic serum concentration [37]. Thereby, the route of administration is the key factor for ameliorating drug bioavailability. In support of this statement, Wong et al. showed that despite griseofulvin being encapsulated within fast-dissolving microparticles, the formulation failed to enhance the drug bioavailability when it was administrated orally through oral gavage technique [74]. The nanosized drug particles adhesiveness feature could also prolong the residence time of the drug on the mucosal surfaces along with the gastrointestinal tract, providing more time for drug absorption and reducing erratic and variable absorption [42]. The nanosized nanocrystalline cellulose-based orally-dispersible film was also responsible for the rapid release, absorption, and enhanced donepezil bioavailability in other literature [75]. Furthermore, the incorporation of tween^®^80 in the oral film formulation might increase tadalafil permeability, owing to its ability to emulsify dispersion and to interact with the mucosal surface to form mixed surfactant-membrane micelles, thereby boosting the drug absorption fraction [76].

### 3.7. Stability

The optimized tadalafil-loaded niosomal film was stable when stored under ordinary and accelerated conditions, where a non-significant alteration in thickness and weight of the film was observed (Table 8). The content of tadalafil was fairly stable ranging from 97.82 ± 0.33% at the beginning to 95.45 ± 2.13% and 94.89 ± 2.34% at the end of the experiment for ordinary and accelerated conditions, respectively (Figure 6). Similar outcomes were observed by Nishimura et al. were the content of prochlorperazine in the oral disintegrating film containing microcrystalline cellulose, polyethlene glycol, and hydroxypropylmethyl cellulose as the polymeric materials was almost constant regardless of storage conditions [77].

## 4. Conclusions

A new generation of tadalafil-loaded surfactant-based vesicle was prepared by ether injection method, optimized using the Box–Behnken Design, and showed unique characteristics such as small vesicular size, unimodal size distribution, and efficient zeta potential. The prepared niosomal dispersions were then successfully loaded into oral polymeric film composed of methylcellulose. The optimized film showed acceptable mechanico-physical characteristics including neutral pH, low moisture content, uniform drug distribution, rapid disintegration, and accepted tensile strength. The film showed a smooth and homogenous surface structure signifying the successful incorporation of the selected niosomal formulation into the polymeric film. Niosomal film displayed also a rapid release of tadalafil within 5 min, which was statistically higher than that of the marketed tablet. Moreover, the in-vivo bioavailability evaluations in rats clarified that the optimized niosomal oral film augmented tadalafil systemic absorption and increased its maximum concentration significantly in comparison to the market tablet. Therefore, polymeric oral film loaded with niosomal tadalafil formulation can be a convenient and economical approach to boost tadalafil absorption and represents a palatable and stable dosage method that can be easily administrated by geriatric patients who suffer from benign prostatic hyperplasia.

## Figures and Tables

**Figure 1 pharmaceutics-15-00173-f001:**
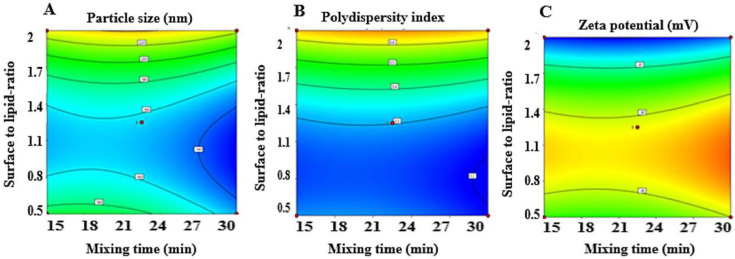
The contour plots of the effect of mixing time (X1) and surfactant to lipid ratio (X2) on particle size (**A**), polydispersity index (**B**), and Zeta potential (**C**) of tadalafil-loaded niosomes. (The red points are the design points).

**Figure 2 pharmaceutics-15-00173-f002:**
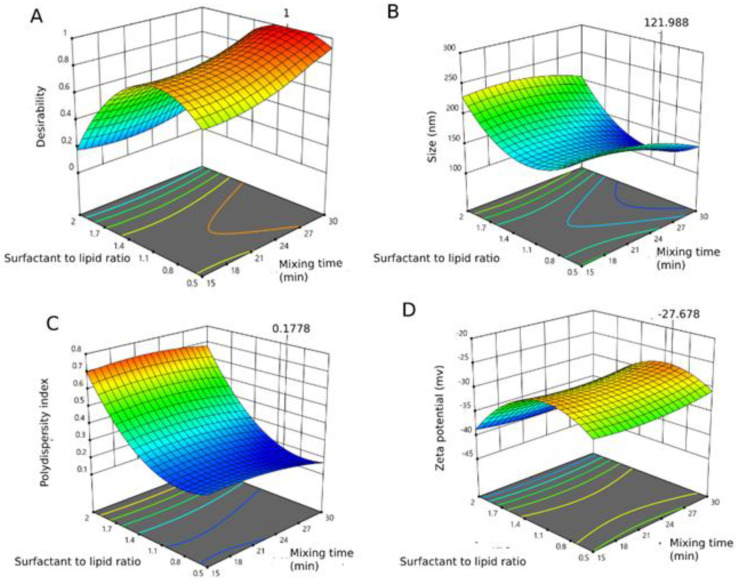
The three-dimensional surface of the desirability function at the factor space (**A**). Individual response is graphed to show the optimum point. Particle size (nm) (**B**), polydispersity index (**C**) and zeta potential (mV) (**D**).

**Figure 3 pharmaceutics-15-00173-f003:**
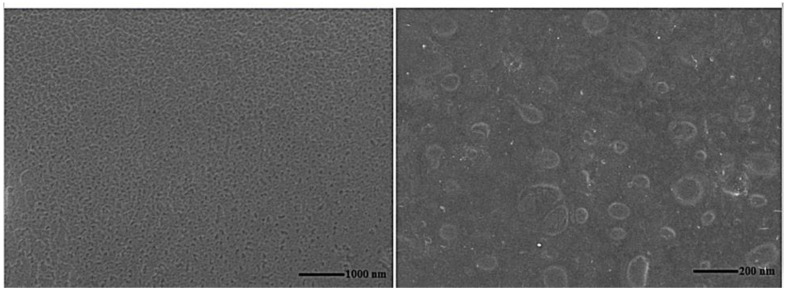
Scanning electron microscope images of oral tadalafil niosomal film.

**Figure 4 pharmaceutics-15-00173-f004:**
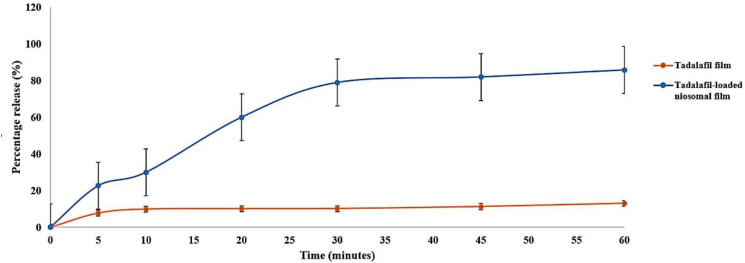
In-vitro percentage release (%) of tadalafil from the niosomal film in comparison with its release from the marketed tablet as function of time (minutes).

**Figure 5 pharmaceutics-15-00173-f005:**
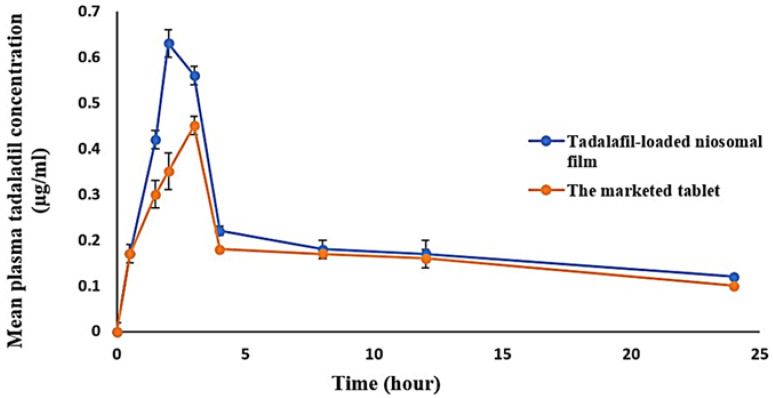
Mean tadalafil plasma concentration versus time after single dose administration of the optimized niosomal oral film and the marketed tablet in two groups of rats.

**Figure 6 pharmaceutics-15-00173-f006:**
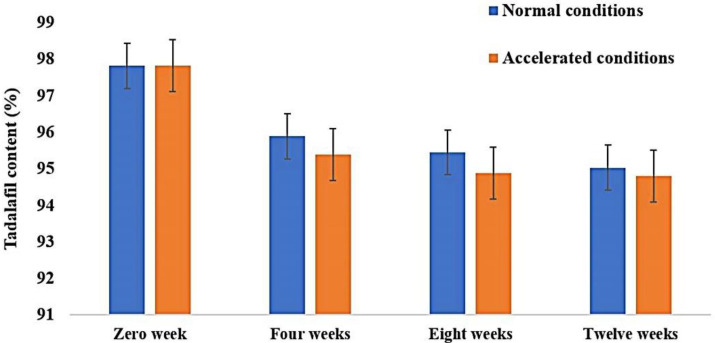
Tadalafil content in the oral niosomal film after storage under different conditions. Each value represents mean ± SD (n = 3).

**Table 1 pharmaceutics-15-00173-t001:** Independent and dependent variables in Box–Behnken design for the optimization of Tadalafil-loaded niosomes.

Independent Variables	Levels		
	Low	Medium	High
X_1_ = Mixing time (min)	15	22.5	30
X_2_ = Non-ionic surfactant to lipid ratio (*w*/*w*)	1:2	1:0.8	2:1
X_3_ = Total weight of the preparation (mg)	200	350	500
Transformed values	−1	0	+1
**Studied Responses**	**Goal**		
Y_1_ = Particle size (nm) Y_2_ = Polydispersity index Y_3_ = Zeta potential (mV)	Minimize Minimize Maximize		

**Table 2 pharmaceutics-15-00173-t002:** Pre-formulation non-ionic surfactants screening.

Surfactant	Particle Size (nm)	Polydispersity Index	Zeta Potential (mV)
Tween^®^80	147 ± 2.63	0.331 ± 0.022	−31.3 ± 0.87
Span^®^60	340 ± 12.06	0.395 ± 0.033	−8.2 ± 0.425
Poloxamer^®^407	217.1 ± 8.20	0.769 ± 0.088	−39.4 ± 0.624

The results are expressed as ± standard deviation (n = 3).

**Table 3 pharmaceutics-15-00173-t003:** Regression analysis for particle size (responses Y_1_), polydispersity index (Y_2_), zeta potential (Y_3_) of tadalafil-loaded niosomes.

Response	Mathematical Model	Adequate Precision	R^2^	Adjusted R^2^	Predicted R^2^	SD	%CV *	*p*-Value ****
Y_1_	Quadratic	23.772	0.987	0.964	0.796	7.510	4.13	0.0001
Y_2_	Quadratic	18.724	0.987	0.965	0.801	0.034	8.69	0.0009
Y_3_	Quadratic	19.724	0.988	0.967	0.832	0.865	2.71	0.0001

* Percentage coefficient of variation. ** Significant *p*-value < 0.05.

**Table 4 pharmaceutics-15-00173-t004:** Box-Behnken design with actual values of the variables *.

Formulation (F)	Mixing Time (min)	Surfactant to Lipid Ratio	Total Weight of the Preparation (mg)	Particle Size (nm)	Polydispersity Index	Zeta Potential (mV)
1	30	1:2	350	145.67 ± 2.34	0.236 ± 0.02	−30.76 ± 3.76
2	22.5	1:0.8	350	153.00 ± 2.78	0.31 ± 0.07	−30.00 ± 7.80
3	22.5	2:1	200	223.10 ± 4.77	**0.760** ± 0.04	**−40.00** ± 8.09
4	22.5	1:0.8	350	157.54 ± 1.26	0.3 ± 1.53	−29.00 ± 3.30
5	15	1:0.8	500	150.90 ± 3.11	0.341 ± 0.06	−28.98 ± 5.81
6	22.5	2:1	500	**267.70** ± 2.29	0.650 ± 0.01	−39.87 ± 7.83
7	22.5	1:2	200	183.90 ± 8.56	0.276 ± 0.05	−31.40 ± 9.52
8	15	1:0.8	200	161.50 ± 6.26	0.288 ± 0.16	−27.98 ± 5.73
9	22.5	1:2	500	194.65 ± 4.76	0.343 ± 1.12	−29.99 ± 7.49
10	15	2:1	350	235.87 ± 7.89	0.660 ± 2.45	−38.87 ± 3.98
11	30	2:1	350	221.76 ± 1.12	0.657 ± 3.12	−37.89 ± 6.17
12	15	1:2	350	189.34 ± 9.32	0.278 ± 0.03	−32.56 ± 8.51
13	22.5	1:0.8	350	159.00 ± 5.21	0.301 ± 3.12	−29.90 ± 8.42
14	30	1:0.8	200	**126.80** ± 2.11	**0.224** ± 0.01	**−27.80** ± 2.91
15	30	1:0.8	500	155.90 ± 4.26	0.341 ± 0.09	−25.60 ± 6.59

* The results are expressed as mean ± standard deviation (n = 3). The bold data represent the upper range and the lower range for each studied parameter.

**Table 5 pharmaceutics-15-00173-t005:** Predicted and observed values of the physicochemical characteristics of the optimized tadalafil-loaded niosomes.

Factor			Optimized Level
X_1_ = Mixing time (min)			29.42
X_2_ = Surfactant: Lipid (ratio)			0.93
X_3_ = Total weight (mg)			243.9
**Response**	**Expected**	**Observed**	**Residual**
Y_1_ = Particle size (nm)	122	126.8	4.8
Y_2_ = Polydispersity index	0.178	0.224	0.046
Y_3_ = Zeta potential (mV)	−27.68	−27.80	0.12

**Table 6 pharmaceutics-15-00173-t006:** Physical and mechanical features of tadalafil-loaded niosomal oral film.

Test	Result
Appearance	Transparent and homogenous
Thickness (µm)	110 ± 10 *
Weight variation (mg)	5.74 ± 0.29 **
Content uniformity (*w*/*w*%)	97.82 ± 0.33 **
Surface pH	6.67 ± 0.49 *
Moisture content (%)	3.02 ± 0.60 *
Folding endurance	320 ± 26.47 *
Tensile strength (MPa)	0.079 ± 0.03 *
Elongation (%)	6.66 ± 0.12 *
Disintegration time (seconds)	30.27 ± 5.06 *

The results are expressed as mean ± standard deviation (* n = 3 and ** n = 10).

**Table 7 pharmaceutics-15-00173-t007:** Tadalafil pharmacokinetics parameters after niosomal oral film and marketed tablet administration.

Pharmacokinetic Parameters *	Oral Niosomal Film	Marketed Tablet	*p*-Value
C_max_ (µg/mL)	0.63 ± 0.03	0.45 ± 0.04	0.001
AUC_0–24_ (µg.h/mL)	4.82 ± 0.51	4.07 ± 0.26	0.01
AUC_0–∞_ (µg.h/mL)	8.90 ± 1.54	7.02 ± 0.83	0.009
K elimination (h^−1^)	0.029 ± 0.001	0.034 ± 0.001	0.05
F	0.71	0.63	0.05
T_max_ (h)	2 ± 0.5	3 ± 0.5	0.0001
t_1/2_ (h)	23.87 ± 4.1	20.05 ± 3.2	0.0003
Relative bioavailability (%)	118.4		

* Data are the mean value ± standard deviation (n = 4).

**Table 8 pharmaceutics-15-00173-t008:** Assessment of stability parameters; thickness and weight of tadalafil-loaded oral niosomal film.

Thickness (µm) *					Weight (mg) *			
Conditions	Zero Week	4 Weeks	8 Weeks	12 Weeks	Zero Week	4 Weeks	8 Weeks	12 Weeks
Ordinary	110 ± 10	106.45 ± 11.32	105.89 ± 3.43	104.44 ± 2.01	5.74 ± 0.29	5.7 ± 1.98	5.66 ± 1.02	5.59 ± 1.89
Accelerated	110 ± 10	105.65 ± 08.35	104.78 ± 7.67	104.18 ± 1.95	5.74 ± 0.29	5.64 ± 3.99	5.34 ± 4.98	5.31 ± 2.94

* Each value was assessed as mean ± SD (n = 3).

## Data Availability

The data presented in this study are available upon request from the corresponding author.

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
