# Peer review of "Application of Box-Behnken Design in the Preparation, Optimization, and In-Vivo Pharmacokinetic Evaluation of Oral Tadalafil-Loaded Niosomal Film"

_pharmaceutics, 2023, doi:10.3390/pharmaceutics15010173_

Round 1
Reviewer 1 Report
Dear Authors,
Please find attached the comments

Author Response
Dear Sir,
Thank you for your e-mail of December the 13rd and for sending on to us the comments of the reviewers of our manuscript ID: 2074389. We have examined these comments carefully and, in their light, have made significant amendments to our manuscript, which are highlighted within the manuscript body. We believe that this has resulted in an improved manuscript and we are grateful to the referees for the comments that they have provided to us. Our point-by-point rebuttal addressing the reviewers’ critiques is enclosed.
For Reviewer #1
We appreciate the reviewer for your valuable and constructive comments and suggestions, which greatly helped us to improve the manuscript. In the light of your comments, we revised our manuscript as follows:
- The authors mentioned that Tadalafil was the first approved PDE-5 inhibitor by US Food and Drug Administration (FDA) for the treatment of erectile dysfunction (ED) and in 2011, but according to many literatures, Sildenafil (Viagra; Pfizer, New York, NY, USA) was the first PDE5 inhibitor approved for treatment of ED in the US in 1998 and Taiwan in 1999.
Response: We thank the reviewer for this invaluable comment. Accordingly, the paragraph was just missing some words as follows:
Tadalafil was approved in 2003 by US Food and Drug Administration (FDA) as the first and only PDE5 inhibitor clinically proven to provide sustained efficacy.
- The authors mentioned that the niosomal approach to enhance tadalafil solubility has not been studied before, despite that a similar work done by the same research group entitled “Tadalafil-Loaded Limonene-Based orodispersible Tablets: Formulation, in vitro Characterization and in vivo Appraisal of Gastroprotective Activity”, with the same aim to enhance tadalafil solubility and finally formulate the optimized formula as orodispersible tables.
Response: As I always say in my research group, tadalafil is still good piece of gold, we already published many articles related to it and our main challenge point was to enhance its solubility, being a class II BCS, one of our papers is already cited as we mentioned in the manuscript but with different aim and rationale, here we use the low dose of it for treatment of BPH while the cited paper, we use an off label use of it as gastroprotective drug against peptic ulcer, and the common point was how to enhance its solubility by different nano-techniques to improve bioavailability and patient compliance
- In the same above point, what is the difference between orodispersible tablets and orodispersible film, and why the authors in the current work select to formulate tadalafil in film while in their recent published work formulate it as tablets
Response: Orodispersible films have advantages over orally disintegrating tablet formulations, which may require more complicated and expensive manufacturing processes, have issues of hardness and friability during manufacturing, storage, handling and administration. In our recent published work, as the reviewer asked, Tadalafil-loaded limonene-based SNES was prepared, and the optimum formula was characterized then loaded on various porous carriers to formulate lyophilized orodispersible tablets (ODTs). Selection of ODTs is much more suitable and serve as a promising revenue for our aim.
- References need to be updated, by simple screening, I found that only 2 out of 76 in 2022 and 7 out of 76 in 2021. This is a critical point need to be taken in consideration.
Also, I did not find reference 21 in the manuscript context, although it is mentioned in reference section.
Response: We apologize for this mistake about missing reference 21, it is now added and more recent references were also added as suggested
- According to my point of view, I feel that authors intend to use some sophisticated non-scientific words throughout the manuscript. I suggest to simplify their words and revise the context of the manuscript by a native English speaker with a pharmaceutical background to make the flow of context easier and clear for all readers.
Response: We disagree with the reviewer in this point, all our words are already scientific and regularly used in literatures, but as per reviewer request, the manuscript was revised by a native English colleague, hope to address the above comment and become more clear to readers.
- The authors used the term “soybean phospholipid” and “Lipoid” interchangeably throughout the context, kindly, unify and use only one.
Response: Corrected and unified in the revised manuscript
- The response equation mentioned in line 130 is not clear, it need to be simply clarified either in methodology or discussion section.
Response: It is a general equation generated by design expert software® to describe the quadratic model, but as per reviewer request, to prevent any confusion to readers, the equation has been omitted in the revised manuscript and renumbering of other equation has been done.
- The authors need to specify clearly what are dependent variables selected in Box–Behnken design for the optimization of tadalafil-loaded niosomes.
Response: The independent variables selected were mixing time (X1), non-ionic surfactant to lipid ratio (X2), and the total weight of the preparation components (X3) with their low, medium, and high levels as dependent value. Kindly refer to page 3, line 125 for the highlighted modification.
- In the preparation method of tadalafil-loaded niosomal oral film, I have two critical points:
- First, the authors mentioned that MC was dispersed in distilled water, followed by the addition of saccharine and menthol, it is well known that MC is simply insoluble in cold water and it need mixture of hot and cold water to be dispersed with no lumps formation in water.
- Response: We thanks the reviewer for this valuable point, MC is soluble in a mixture of hot and cold water in a ratio 2:1 by solvent casting method as mentioned in reference 26 with slight modification. Kindly refer to page 4, line 156 for the highlighted modification.
- Second, I have some concern about drying method, it seems that it is a simple air drying method, what is your ending point, how can you judge the complete drying and what about the thickness and spreadability and smoothness of the prepared films.
- Response: As mentioned above, preparation method was done with slight modification according to reference 26. The solution was poured on the petri dish (surface area, 19.63 cm2) and allowed to dry at room temperature until a constant weight, the latter words were now added in the revised manuscript (Page 4, line 161)
- In the section of in vitro release studies, what is the volume of dissolution medium and why the authors used only 0.1N HCl as dissolution medium, in the oral cavity, the pH is maintained near neutrality (6.4-6.8) by saliva and it is logic that fast dissolving film will be dissolved in saliva not GIT, so the choice of 0.1N HCl as dissolution medium may be wrong choice.
Response: The volume of dissolution medium was 100 mL, it is now mentioned in revised manuscript. Concerning the second point, as stated in methodology, solution of 0.1N HCl and 2% w/v tween®80 (pH 1.2) was used as a dissolution medium as it was previously proved in literature to be a compatible medium for the release of tadalafil (Reference 14).
- The authors compare between the prepared oral film and the marketed tablet in the pharmacokinetics section, is that a good choice of comparison to compare between tablet and film and what about the tablet itself is that a fast dissolving one or simple core tablet. This is another critical point related to the core of the manuscript need to be clarified.
Response: We thanks the reviewer for this valuable comment. Yes, as you expected, marketed tablets were fast dissolving which make our work rationale with in the right track.
- In stability studies, the authors select to screen only the content of tadalafil, weight and thickness. I see that the most critical parameter that was need to be studied is the in vitro release.
Response: We agree with the reviewer but unfortunately, in vitro release studies have been done in stability work in one run not as in triplicate so we were not honestly add results in manuscript, but that was a good guide for stability. Similar stability studies conditions, without in vitro release one, were already done in cited reference (Nishimura, M etal 2009) as mentioned in the manuscript.
- What is the symbol (*) meaning in equations 6-8?
Response: It is a multiplication symbol. To avoid confusion, it is now omitted in revised manuscript.
- In Table 3, what is the difference between R2, adjusted R2 and predicted R2, the authors mentioned that the difference between adjusted and predicted R2 values for the investigated responses was less than 0.2 which indicates a reasonable agreement in the study design, please clarify.
Response: It is well known in statistical studies that R2 and Adjusted R2 give an explanation of how well your dependent variable is explained by your independent variables. It has nothing to do with the predictive quality of your model. Predicted R2 measures the predictive quality of your model. That is the ability of the model to predict a set of new data. The agreement with selected study design is achieved when difference between adjusted and predicted R2 values for the investigated responses is lower as possible, here in our design, the difference was less than 0.2 which indicates a reasonable agreement.
- Does the authors compare between different mathematical models (linear, 2FI and quadratic) to judge and select the quadratic one as mentioned in Table 3.
Response: Yes, a comparison between different mathematical models (linear, 2FI and quadratic) to judge the obtained data was done by design expert software and it proposed a quadratic model for the analysis of selected responses (particle size, PDI, and zeta potential) in the study design.
- There is 15 formulae not 27, it is meaning incomplete or partial design, also, there is three identical formulae with nearly the same response (F2, F4 and F13), I need more explanation for this point.
Response: Box-Behnken Design is an incomplete or partial factorial design that was selected to decrease number of formulae.
Concerning the identical formulae, they are well known in design expert as central points. Replicated center points are used to estimate pure error for the lack of fit test. Lack of fit indicates how well the model you have chosen fits the data. With fewer than five or six replicates, the lack of fit test has very low power
- In Figure 1, the author stated that red points are the design points, what does it meaning?
Response: They are the selected central points, kindly, refer to explanation mentioned in previous question.
- In line 362, the authors mentioned that F14 was the selected formula, does it mean that the future experiments were done on F14 or on the optimized formula present in Table 5.
Response: formula (F14) with the smallest particle size (126.80 nm), lowest PDI (0.224), and optimal zeta potential (-27.80 mV) was selected to be the most successful one when comparing with all 15 formulae in the experimental design. But, for future experiments and niosomal film formulation and characterization, we use the optimized formula created by the experimental design. Kindly, refer to Table for more details.
- I just wondering, how can you calculate the standard deviation in folding endurance experiment as mentioned in table 6.
Response: The mechanical strength was determined through the folding endurance of the pre-pared films which was computed based on the number of times the film can fold at the same place without breaking as reported by Kathpalia, H etal (Reference 36). The experiment was done on three niosomal films, mean and standard deviation were simply calculated. Kindly back to Table 6 where number of replications were already mentioned.
As the film tolerates more than 300 times folding, the more mechanical strength of the film. The folding endurance of tadalafil film was about 320 indicating good flexibility.
- The authors stated that “the disintegration time ranges from 5 to 30 seconds to allow faster drug release and fast oromucosal absorption”. Does it mean that Tadalafil will be completely absorbed in mouth, if so, it will be like sublingual one or buccal film/tablet and that will completely change the concept of the work?
Response: We thanks the reviewer for the valuable comment, it is a typo-mistake, we still stick to our work rationale to formulate and assess oral film containing tadalafil-loaded niosomes to provide a fast systemic effect by improving tadalafil solubility through a patient-geriatric friendly formulation.
- Figure 3 is not clear, kindly replace with another one, if present, to clarify the results.
Response: The Figures presented in the word document form reflected poor image quality however; the figures that are attached separately meet the journal requirements with enhanced resolution.
- Figure 4, In-vitro percentage release of tadalafil from the niosomal film in comparison with its release from the marketed tablet, does the authors use marketed tablet or tadalafil film as stated within the discussion of this part?, I am really confused or is it a typo mistake in figure caption.
Response: We apologize for this typo-mistake. It is now corrected in the revised manuscript to tadalafil film, as already written on the figure itself.
- The authors stated that “the oral niosomal film provided a dual release pattern characterized by a fast-dissolving profile at the beginning of the experiment yielded by the film components and a controlled one for the remaining time assured by niosomes”
I am confused again with the aim of this work, how you can keep the film in mouth for one hour to achieve the controlled release pattern, it look like a mucoadhesive film not a fast dissolving film.
Response: We apologize for such confusion, this paragraph was come in just an earlier location, and it should be stated after the in-vivo pharmacokinetic assessment to make a more reliable discussion, it is now re-allocated in the revised manuscript.
As mentioned in the discussion section, the presence of tadalafil within the niosomes as a nanocarrier enhanced its solubility as well as its dissolution rate compared to the marketed product by rapid disintegration of the niosomal oral film in the saliva with faster tadalafil absorption via the buccal mucosa, in addition to, the nanosized drug particles adhesiveness feature could also prolong the residence time of the drug on the GIT mucosal surfaces, providing more time for drug absorption and reducing erratic and variable absorption.
We hope the reviewer could accept our Explanation
We believe that our revised manuscript reaches the criteria to accept for publication.
Finally, we really thank the reviewer for kind and critical comments.

Reviewer 2 Report
The authors demonstrated the oral niosomal film containing Tadalafil using Box-Behnken Design to obtain optimal ratio of niosomes film composition. Moreover, the pharmacokinetics characteristics were appraised in rats against the marketed product. This manuscript provided important insight on controlled delivery system using vesicular drug delivery. However, some major and minor changes would strengthen the manuscript.
1. The authors successfully developed the oral niosomal film. Due to the oral taken, niosomes containing tween80, lipoid S75 and Tadalafil, these compositions may cause bitter taste. Do the authors concern about this point.
2. For the application of oral film, which route of drug administration the authors intend to apply, sublingual or buccal or?
3.For the Entrapment efficiency, please elaborate more about the method of analysis. Moreover, the authors mention about the subtraction of the total drug and the free drug, but the equation showed only the entrapped drug. In my opinion, I think the method that the authors mentioned is indirect method. If it is not, please explained.
4. For the content uniformity, the authors used UV spectrophotometric at 291 nm. Do the other compositions disturb this wavelength? Please verify.
5. For the in vivo pharmacokinetics section.
5.1 How the authors obtain the 5 rats per group. Dose the authors use sample size calculation? Which program and reference that the authors use or cite?
5.2 The authors devised the experiment into 2 groups. Group 1 the oral film and group 2 the commercial tablet. Please explain why the group 1 given the intact film while group 2 given oral suspension. Please provide the criteria or reason why the group 1 did not dissolve the film and gave the extract?
5.3 How does the authors obtain the relative bioavailability? Please add more information this topic and provide the equation for the relative bioavailability?
5.4 For the bioavailability, does the author perform any experiment to obtain the fraction of absorption (F)?
6. The stability study was performed at two temperatures as one accelerated temperature (40°C) and at 25°C, but it should be performed for at least 3 or 6-months.
7. In Table 4, why did the authors use the bold letters on some data as F3, F6 and F14. If the authors would like to use in order to show the difference, please elaborate more on the Table caption.
8. In figure 4, the graph showed that the %release of oral film is quite low compared to the niosomal film. Do the authors perform biodegradable study along with the drug release?
9. Ether is an organic solvent that can cause toxicity. Do the authors perform the experiment to measure the ether residual left on the film?
Minor revision
1. Please check the format on MDPI website. For example, at the conclusion paragraph, please used justify.
2. Please check the references format. For example, at reference number 17.

Author Response
Dear Sir,
Thank you for your e-mail of December the 13rd and for sending on to us the comments of the reviewers of our manuscript ID: 2074389. We have examined these comments carefully and, in their light, have made significant amendments to our manuscript, which are highlighted within the manuscript body. We believe that this has resulted in an improved manuscript and we are grateful to the referees for the comments that they have provided to us. Our point-by-point rebuttal addressing the reviewers’ critiques is enclosed.
For Reviewer #2
We appreciate the reviewer for your valuable and constructive comments and suggestions, which greatly helped us to improve the manuscript. In the light of your comments, we revised our manuscript as follows:
- “The authors successfully developed the oral niosomal film. Due to the oral taken, niosomes containing tween 80, lipoid S75 and Tadalafil, these compositions may cause bitter taste. Do the authors concern about this point?”
Response: As this niosomal film was intended for oral administration, menthol and saccharine were incorporated in the preparation as flavoring and sweeting agent, respectively. Kindly check page 4 line 153 “saccharine as a sweetener, and menthol as a flavoring agent.”
- “For the application of oral film, which route of drug administration the authors intend to apply, sublingual or buccal or?”
Response: The authors intend to apply the oral film through the buccal site since it is a promising area for systemic drug delivery owing to its features in overcoming the problem of first-pass drug metabolism, which enhances the drug bioavailability, reduces the dose, and consequently reduces the side effect. In addition, drug toxicity can be promptly terminated by removing the dosage from the buccal cavity. It is also possible to administer drugs to patients who cannot be dosed orally. The route of administration has been also cleared in the pharmacokinetic section. Please refer to page 5 line 246
- “For the Entrapment efficiency, please elaborate more about the method of analysis. Moreover, the authors mention about the subtraction of the total drug and the free drug, but the equation showed only the entrapped drug. In my opinion, I think the method that the authors mentioned is indirect method. If it is not, please explained the.”
Response: The entrapment efficiency mentioned followed the indirect method as presented in the article, the authors did subtract the total amount of drug from the free drug (kindly check page 4 line 145 and 146). This was edited in the article for more clearance as per the reviewers’ request. Kindly check page 4 line 142 and equation 2.
- “For the content uniformity, the authors used UV spectrophotometric at 291 nm. Do the other compositions disturb this wavelength? Please verify”
Response: Based on literature, the following compounds have the respective wavelengths, and none of them interfere with that of tadalafil.
|
Ingredient |
Wavelength |
|
Microcrystalline cellulose |
330 nm |
|
Polyethylene glycol |
Weak and broad absorption peak at 420 to 440 nm |
|
Tween®80 |
234 nm |
|
Saccharine |
267.3 nm |
|
Menthol |
498 nm |
|
Tadalafil |
291 nm |
- For the in vivo pharmacokinetics section.
5.1. “How the authors obtain the 5 rats per group. Dose the authors use sample size calculation? Which program and reference that the authors use or cite?”
Response: The authors used the minimal number of animals sufficient for statistically significant testing and in line with the animal right and ethical considerations and our IRB approval. The following references have also utilized the same number of rats in the in-vivo pharmacokinetic study (n=4).1,2
- Wong SM, Kellaway IW, Murdan S. Fast-dissolving microparticles fail to show improved oral bioavailability. J Pharm Pharmacol. 2010;58(10):1319-1326. doi:10.1211/jpp.58.10.0004
- Anjireddy K, Karpagam S. Micro and nanocrystalline cellulose based oral dispersible film; preparation and evaluation of in vitro/in vivo rapid release studies for donepezil. Brazilian J Pharm Sci. 2020;56:1-17. doi:10.1590/s2175-97902020000117797
5.2: “The authors devised the experiment into 2 groups. Group 1 the oral film and group 2 the commercial tablet. Please explain why group 1 given the intact film while group 2 given oral suspension. Please provide the criteria or reason why the group 1 did not dissolve the film and gave the extract?”
Response: The authors aimed in the current work not only to improve tadalafil solubility and bioavailability through the nanocarriers but also to develop a patient-geriatric friendly formulation. Hence, the first group received an oral film to improve this that formulation which has a good patient compliance is also able to ameliorate tadalafil systemic profile comparing with the oral tablet.
5.3: “How does the authors obtain the relative bioavailability? Please add more information this topic and provide the equation for the relative bioavailability?”
Response: The relative bioavailability was calculated according to the following equation:
RB
RB= (4.82/4.07)* 100=118.4 %
The equation was added. Please check page 6 line 279
5.4: “For the bioavailability, does the author perform any experiment to obtain the fraction of absorption (F)?”
Response: In the case of oral absorption, F is usually defined based on the total amount of a drug that reached the systemic circulation after oral administration. It can be calculated according to the following equation:
F*x = AUC * CL
Where,
F is the fraction of drug absorbed
X is the oral dose
AUC is the area under the curve
CL is the clearance
So according to the obtained data
F for the oral film is 0.7
F for the oral tablet is 0.63
The obtained new data were added to the table 7, please check page 15
- “. The stability study was performed at two temperatures as one accelerated temperature (40°C) and at 25°C, but it should be performed for at least 3 or 6-months.”
Response: The stability study was performed for three months, however there was no further change after week 8 in the studied parameters so the data was only written for the first 8 weeks.
- “In Table 4, why did the authors use the bold letters on some data as F3, F6 and F14. If the authors would like to use in order to show the difference, please elaborate more on the Table caption.”
Response: The bold data in Table 4 were meant to show the upper range and the lower of each studied parameter, kindly check page 7 lines 318-320 “The particle size ranged from 126.80 nm (F14) and 267.70 nm (F6), polydispersity index from 0.224 (F14) and 0.760 (F3) and the zeta potential from -27.80 mV (F14) and -40.00 mV (F3) (Table 4)”. Additionally, and for more clearance for the readers Table 4 caption was edited for more elaboration. Kind check page 8 lines 329-330 for the changes.
- “In figure 4, the graph showed that the %release of oral film is quite low compared to the niosomal film. Do the authors perform biodegradable study along with the drug release?”
Response: Before performing the in-vitro release study, the authors have studied the effect of the dissolution medium consisting of a solution of 0.1N HCl and 2% w/v tween®80 (pH 1.2) on the disintegration of the niosomal film. The film was able to break down within about 2 minutes after immersing in the dissolution medium, thus, the effect of the film on the release results is considered null.
- “Ether is an organic solvent that can cause toxicity. Do the authors perform the experiment to measure the ether residual left on the film?”
Response: Before preparing the film, the residual ether was tested to detect if there is any remaining ether after evaporation.
The test was performed as following; 2 mL of the niosomes were set to boil in a tube cover with a filter paper moistened with a mixture of cupric acetate and benzidine hydrochloride solution. Upon boiling the appearance of a deep blue color indicated the presence of ether, and in this case the niosomes were set for more heating, this was repeated until no further blue color appear.
Minor revision
- Please check the format on MDPI website. For example, at the conclusion paragraph, please used justify.
Response: The conclusion paragraph was justified as requested, check page 16.
- Please check the references format. For example, at reference number 17.
Response: Reference 17 was corrected as per the journal format, kindly check page 18.
We hope the reviewer could accept our Explanation
We believe that our revised manuscript reaches the criteria to accept for publication.
Finally, we really thank the reviewer for kind and critical comments.

Round 2
Reviewer 1 Report
The authors addressed all the raised points and the manuscript is legible to be published now.
Author Response
We really thank the reviewer for kind comments.

Reviewer 2 Report
The authors explained and revised all of the comments to make this manuscript more scientific. Some minor points should be corrected before publication.
1. Please kindly check the equation and rearrange all equation. For example,
in text line 148 page 4, should be equation 1 not equation 2. Moreover, please add the equation number for Fraction of absorption and relative bioavailability equation.
2. If the authors cite or perform following this references
1. Wong SM, Kellaway IW, Murdan S. Fast-dissolving microparticles fail to show improved oral bioavailability. J Pharm Pharmacol. 2010;58(10):1319-1326. doi:10.1211/jpp.58.10.0004 2. Anjireddy K, Karpagam S. Micro and nanocrystalline cellulose based oral dispersible film; preparation and evaluation of in vitro/in vivo rapid release studies for donepezil. Brazilian J Pharm Sci. 2020;56:1-17. doi:10.1590/s2175-97902020000117797
please cite and credit. If not, please declare.
3. Some suggestion for the stability. If there is no further change after week 8, please provide the recent information or at least 3 months. Only 8 weeks could not be called standard stability performance in term of the stability guideline.
Author Response
Responses to reviewers’ comments
Dear Sir,
Thank you for your e-mail of December the 26th and for sending on to us the comments of the reviewers of our manuscript ID: 2074389. We have examined these comments carefully and, in their light, have made significant amendments to our manuscript, which are highlighted within the manuscript body. We believe that this has resulted in an improved manuscript and we are grateful to the referees for the comments that they have provided to us. Our point-by-point rebuttal addressing the reviewers’ critiques is enclosed.
Reviewer 2
The authors explained and revised all of the comments to make this manuscript more scientific. Some minor points should be corrected before publication.
- Please kindly check the equation and rearrange all equation. For example, in text line 148 page 4, should be equation 1 not equation 2. Moreover, please add the equation number for Fraction of absorption and relative bioavailability equation.
- Response: Kindly check lines 150, 194, 204, 207, 282, 284, and 320.
- If the authors cite or perform following this references
- Wong SM, Kellaway IW, Murdan S. Fast-dissolving microparticles fail to show improved oral bioavailability. J Pharm Pharmacol. 2010;58(10):1319-1326. doi:10.1211/jpp.58.10.0004 2. Anjireddy K, Karpagam S. Micro and nanocrystalline cellulose based oral dispersible film; preparation and evaluation of in vitro/in vivo rapid release studies for donepezil. Brazilian J Pharm Sci. 2020; 56:1-17. doi:10.1590/s2175-97902020000117797
Please cite and credit. If not, please declare.
- Response: The authors have compared the obtained in-vivo results with the outcomes of the above-mentioned references, thus, they were cited. Please check lines: 586 and 592
- Some suggestion for the stability. If there is no further change after week 8, please provide the recent information or at least 3 months. Only 8 weeks could not be called standard stability performance in term of the stability guideline.
- Response: The data of the 12th week were added to table 8 and figure 6. Kindly check lines 624 and 628
We hope the reviewer could accept our Explanation
We believe that our revised manuscript reaches the criteria to accept for publication.
Finally, we really thank the reviewer for kind and critical comments.
